# Human RPF1 and ESF1 in Pre-rRNA Processing and the Assembly of Pre-Ribosomal Particles: A Functional Study

**DOI:** 10.3390/cells13040326

**Published:** 2024-02-10

**Authors:** Alexander Deryabin, Anastasiia Moraleva, Kira Dobrochaeva, Diana Kovaleva, Maria Rubtsova, Olga Dontsova, Yury Rubtsov

**Affiliations:** 1Shemyakin-Ovchinnikov Institute of Bioorganic Chemistry RAS, 119997 Moscow, Russia; 2Department of Applied Mathematics, MIREA-Russian Technological University, 119454 Moscow, Russia; 3Department of Chemistry, Lomonosov Moscow State University, 119991 Moscow, Russia; 4N.N.Blokhin National Medical Research Center of Oncology, Ministry of Health of the Russian Federation, 115478 Moscow, Russia

**Keywords:** ribosome biogenesis, nucleolus, ribosomal RNA processing

## Abstract

Ribosome biogenesis is essential for the functioning of living cells. In higher eukaryotes, this multistep process is tightly controlled and involves a variety of specialized proteins and RNAs. This pool of so-called ribosome biogenesis factors includes diverse proteins with enzymatic and structural functions. Some of them have homologs in yeast *S. cerevisiae,* and their function can be inferred from the structural and biochemical data obtained for the yeast counterparts. The functions of human proteins RPF1 and ESF1 remain largely unclear, although RPF1 has been recently shown to participate in 60S biogenesis. Both proteins have drawn our attention since they contribute to the early stages of ribosome biogenesis, which are far less studied than the later stages. In this study, we employed the loss-of-function shRNA/siRNA-based approach to the human cell line HEK293 to determine the role of RPF1 and ESF1 in ribosome biogenesis. Downregulating RPF1 and ESF1 significantly changed the pattern of RNA products derived from 47S pre-rRNA. Our findings demonstrate that RPF1 and ESF1 are associated with different pre-ribosomal particles, pre-60S, and pre-40S particles, respectively. Our results allow for speculation about the particular steps of pre-rRNA processing, which highly rely on the RPF1 and ESF1 functions. We suggest that both factors are not directly involved in pre-rRNA cleavage but rather help pre-rRNA to acquire the conformation favoring its cleavage.

## 1. Introduction

Ribosome biogenesis is a sophisticated time-ordered process, which adjusts the protein synthesis rate for the consumption of nutrients and external stimuli. It begins with the transcription of the ribosomal primary RNA precursor (35S/47S in *S. cerevisiae/H. sapiens*). 47S pre-rRNA processing (Figure 1) is coupled with the sequential recruitment of ribosome biogenesis factors and non-coding RNAs as well as the ordered coating of rRNA with ~80 ribosomal proteins during the formation of the functional 60S and 40S ribosomal subunits [1]. Despite the considerable progress in understanding the ribosomal biogenesis in lower eukaryotes, in particular that in yeast *S. cerevisiae* [2,3,4,5,6,7,8], the complexity of the nucleolus and ribosomal precursors in higher eukaryotes hinders progress in obtaining structural and functional data on ribosome biogenesis [9,10,11]. 47S pre-rRNA processing involves multiple human proteins that do not share sequence homology with *S. cerevisiae*, thereby implying that the biogenesis of ribosomal subunits in humans is more complex and most likely involves steps that still remain poorly understood [12].

Rpf1 and Esf1 are the nucleolar factors contributing to the biogenesis of 60S and 40S in *S. cerevisiae*, respectively. Both have human homologs, which are RPF1 and ESF1, respectively [13,14]. A recent Cryo-EM study revealed the arrangement of RPF1 in the structure of human 60S subunit precursors. According to the structural data, RPF1 is present in the intermediates A1, B1, C1, and D1 and dissociates upon NVL2 recruiting and association [12]. It is not clear whether it is required for the cleavage steps or provides a structure favoring the change in the pre-rRNA conformation that is mentioned above. Studying RPF1 in yeast uncovered both biochemical and structural details [6,8,13]. RPF1-RNA co-immunoprecipitation experiments demonstrate its association with 35S, 27SA, and 27SB pre-ribosomal RNAs within the ITS1 and ITS2 sites. Rpf1 knockdown in yeast leads to the accumulation of the 27SA3 precursor and a decrease in the 7S precursor level, suggesting its possible interaction with the A3 site in the ITS1 and C2 sites in ITS2. Moreover, Rpf1 is thought to be a nucleation factor of the pre-66S complex in yeast [13]. Less is known about the ESF1 protein. Esf1 knockdown in yeast leads to a dramatic decrease in 27SA2 and 20S pre-rRNAs, implying its role in the A2 site cleavage. Furthermore, the accumulation of the 35S and aberrant 23S pre-ribosomal RNAs has been observed, implicating the possible inhibition of A0 and A1 cleavage [14]. Meanwhile, there are no biochemical or structural data concerning human ESF1 and its contribution to human ribosome biogenesis.

Here, we show that human RPF1 and ESF1 are directly involved in ribosomal subunits maturation. We evaluated the impact of RPF1 and ESF1 loss of function on their subcellular localization, pre-ribosomal RNA profiles, and association with the 60S and 40S precursors. We demonstrate that both RPF1 and ESF1 interact with the precursors of the large and small ribosomal subunits, respectively, and are thus involved in the early steps of their biogenesis.

## 2. Materials and Methods

### 2.1. DNA Constructs and Cloning

The shRNAs-coding sequences against *RPF1* or *ESF1* mRNAs or scramble control were selected from the Sigma MISSION^®^ libraries and ordered as 21-mer top and bottom strands. Oligonucleotides were annealed and cloned into a pLKO.1-TRC vector (Addgene #10878) following the manufacturer’s protocol. The E. coli Stbl3 strain was used to prepare ultra-competent cells according to the Inoue method for routine cloning of pLKO.1-TRC-derived constructs to avoid any recombination events and increase the plasmid yield.

### 2.2. Cell Culture, Lentivirus Particles Preparation, Viral Transduction, and siRNA Transfection

The HEK293 cell line was obtained from the Russian bank of cell lines (Institute of Cytology RAS, Saint Petersburg, Russia). Cells were grown in DMEM (PanEco, Moscow, Russia) and supplemented with 10% HyClone fetal bovine serum (Cytiva, Marlborough, MA, USA), 2 mM of L-glutamine, and penicillin/streptomycin (250 units/mL, all from PanEco, Moscow, Russia) at 37 °C and 5% CO_2_. To produce lentivirus particles, HEK293 cells were co-transfected with plasmids encoding shRNA, GAG/Pol, Rev, and VSV-G using Lipofectamine 2000 (ThermoFisher, Wilmington, DE, USA) according to the protocol (http://www.LentiGO-Vectors.de, accessed on 21 September 2022). The medium containing lentivirus particles was filtered through a 0.45 μm filter and used immediately or frozen and stored at −80 °C until use. Approximately 60% confluent wild-type HEK293 cells were transduced with the obtained lentivirus particles using Polybrene reagent added at a final concentration of 10 μg/mL (Miltenyi Biotec, Bergisch Gladbach, Germany). The pCDH-copGFP plasmid (10% of the total DNA amount) was added to a DNA mix to check the transduction efficiency. A total of 24 h after transduction, the medium was replaced with the fresh one without viral particles. Then, 3 days after transduction, puromycin was added to a final concentration of 10 μg/mL for cell selection. After selection, the knockdown efficiency was analyzed by RT-PCR and Western blotting.

Two siRNAs against *RPF1* mRNA were designed and synthesized with additional modifications to improve their intracellular stability (GenTerra, Moscow, Russia). Approximately 70% confluent wild-type HEK293 cells were transfected by siRNAs individually or together at working concentrations of 5 nM and 15 nM according to Lipofectamine RNAiMAX (ThermoFisher, Wilmington, DE, USA) protocol. mRNA and protein level analysis as well as Northern blots were performed 3 days after transfection.

### 2.3. RT-PCR

Total RNA was isolated from the cells stably expressing shRNAs against *RPF1* or *ESF1* mRNAs or scramble control shRNA using TRIzol (ThermoFisher, Wilmington, DE, USA). RNA quality and concentration were assessed using the NanoDrop OneC spectrophotometer (ThermoFisher, Wilmington, DE, USA) at 260/280 nm wavelengths. cDNAs were prepared from 1 μg of RNA from each sample using SuperScript IV RT (ThermoFisher, Wilmington, DE, USA). The obtained cDNAs were diluted 10-fold and subjected to RT-PCR with primers to *RPF1*, *ESF1*, and *GAPDH* sequences spanning exon–exon junctions using 5X qPCRmix-HS SYBR (Evrogen, Moscow, Russia) according to the vendor’s protocol.

To prepare cDNA from ethynyl uridine-labeled RNA (see the corresponding sections in Materials and Methods), 50 μL of streptavidin magnetic beads with the bound biotin-RNA were incubated with a primer annealing solution (contains mix of primers, RNAse inhibitor, RNAse-free water) at 70 °C for 3 min. Then, the reverse transcriptase (RNAscribe RT, Biolabmix, Moscow, Russia) diluted in a reaction buffer was added, and the final mix was incubated following the manufacturer’s protocol. A total of 2.5 μL of the obtained cDNA were used for further RT-PCR with primers to 5′ETS or *GAPDH*. The primer sequences were as follows: 5′ETS-forward CGCGGGCCTGCTGTTCTCTC, 5′ETS-reverse CCCCGGGTGGGTCAGAGACC, GAPDH-forward GAAGGTGAAGGTCGGAGTCA and GAPDH-reverse TTGAGGTCAATGAAGGGGTC.

### 2.4. SDS-PAGE and Western Blotting

A total of 1 × 10^6^ HEK293-derived modified cells were lysed on ice in 200 μL of the lysis buffer (50 mM of Tris-HCl, 150 mM of NaCl, 10% glycerol, 1% Triton X-100, 1 mM of PMSF, 1 mM of DTT, at pH 7.5). Protein concentration in lysates was determined by Bradford reaction according to the standard protocol. Lysate samples or proteins extracted from sucrose were diluted with 5x Laemmli buffer and subjected to SDS-PAGE (25 μg per lane) in 12% gel prior to transfer to the nitrocellulose membrane (Bio-Rad Laboratories, Hercules, CA, USA). The membrane was blocked in 5% non-fat dry milk in TBST (20 mM of Tris-HCl, 150 mM of NaCl, 0.1% Tween 20, at pH 7.6) for 1 h at room temperature and incubated with rabbit anti-RPF1 (HPA024642, Sigma, Suffolk, UK, 1:1000 dilution), anti-ESF1 (SAB1411774, Sigma, Suffolk, UK, 1:1000 dilution), or anti-actin (A4700, Sigma, Suffolk, UK, 1:1000 dilution) primary antibodies in TBST supplied with 5% non-fat dry milk for 1 h at room temperature. After three 5 min washes in TBST, the membrane was incubated with the secondary anti-rabbit HRP-conjugated IgG (Sigma, Suffolk, UK, 1:20,000 in TBST supplied with 5% non-fat dry milk) for 1 h at room temperature. The membrane was then washed three times in TBST and incubated with ECL reagents (Cytiva, Marlborough, MA, USA) according to the vendor’s instruction and visualized using ChemiDoc imager (Bio-Rad Laboratories, Hercules, CA, USA). Densitometry analysis was performed with ImageLab 5.0 software.

### 2.5. Northern Blotting

Total RNA from HEK293-derived stable cell lines was isolated using TRIzol (ThermoFisher, Wilmington, DE, USA) following the standard protocol. RNA quality and concentration were measured using the NanoDrop OneC spectrophotometer (ThermoFisher, Wilmington, DE, USA) at 260/280 nm wavelengths. RNA was separated by gel electrophoresis in a 1.2% agarose gel containing 1.8 M of formaldehyde in 20 mM of MOPS/4 mM of NaAc/ 0.5 mM of EDTA (pH 7.0) used as a buffer. A total of 3 μg of total RNA were loaded to each lane and run at 2V/cm for 6 h. RNA was transferred to the BrightStar^®^-Plus positively charged nylon membrane (ThermoFisher, Wilmington, DE, USA) for 2 h according to the alkali transfer protocol. A total of 1 M of NaCl/10 mM of NaOH was used as a transfer buffer. The membrane was rinsed in 2x SSC buffer (0.3 M of NaCl, 30 mM of sodium citrate, at pH 7.0) and baked at 80 °C for 15 min in an oven. Hybridization with the biotin-labeled probes (10 ng/mL final concentration) was performed in the ULTRAhyb^®^-Oligo buffer (ThermoFisher, Wilmington, DE, USA) according to the manufacturer’s instructions at 42 °C for 12–16 h. The probe sequences were as follows: 5′ITS2-biotin–GGGGCGATTGATCGGCAAGCGACGCTC, 5′ITS1-biotin–GGCCTCGCCCTCCGGGCTCCG, 7SK-biotin-GACGCACATGGAGCGGTGAGGGAG, 28S-biotin-CCTCTTCGGGGGACGCGCGCGTGGCCCCGA and 18S-biotin-ATCGGCCCGAGGTTACTCTAGAGTCACCAAA. After hybridization, the membrane was washed twice with 2× SSC buffer supplemented with 0.5% SDS for 30 min at 42 °C. Bands were visualized with the Chemiluminescent Nucleic Acid Detection Module (ThermoFisher, Wilmington, DE, USA) at room temperature according to the manufacturer’s protocol using the x-ray film or ChemiDoc Imaging System (Bio-Rad Laboratories, Hercules, CA, USA). Densitometry analysis was performed by ImageLab 5.0 software.

### 2.6. Analysis of rRNA Precursor Ratios

RAMP (Ratio Analysis of Multiple Precursors) profiles were generated according to [15]; the density of rRNA precursor bands was determined as previously described. The ratio of precursors was calculated for each total RNA sample (pre-rRNA 41S/45-47S, 12S/32S, etc.) and was obtained from analyzing each individual lane following the hybridization with one probe. The log2 values of precursor ratios in the control samples were subtracted from log2 values of corresponding ratios in the samples from cells with either RPF1 or ESF1 knockdown. Resulting normalized log2 values were plotted on histograms and combined two different ratios into one RAMP profile. The first one represented the ratio of the corresponding pre-rRNA to the primary rRNA transcript, and the second represented the ratio between pre-rRNAs that formed substrate–product pairs, for example, 18SE/21S and 12S/32S.

### 2.7. Immunocytochemistry

HEK293-derived stable cells were grown on cover slides. Cells were fixed in absolute acetone at −20 °C for 10 min and were incubated with a mixture of two antibodies as follows: rabbit anti-RPF1/anti-ESF1 (1:100 dilution) and either mouse anti-B23/nucleophosmin or mouse anti-SURF6 (1:200 dilution, Sigma, Suffolk, UK, B0556) for 1 h at room temperature. Cells were washed in PBS for 3 × 5 min and stained for 1 h at room temperature with a mixture of AlexaFluor 488 goat anti-mouse IgG (HCL) antibodies (ThermoFisher, Wilmington, DE, USA, A-11034, for detection of B23) and Alexa Fluor^®^ 568 goat anti-rabbit IgG (H+L, ThermoFisher, Wilmington, DE, USA, A11004, for detection of RPF1/ESF1). To study co-localization with SURF6, cells were stained with a mixture of rabbit anti-RPF1/ESF1 and mouse anti-SURF6 antibodies, washed in PBS for 3x5 min, and incubated for 1 h at room temperature with a mixture of goat anti-mouse and goat anti-rabbit antibodies, as described above.

After incubation with secondary antibodies, the cells were washed with PBS for 3 × 5 min, stained with DNA-binding dye DAPI (1 mg/mL in PBS, 10 min) and fixed in Vectashield (Vector Laboratories, Newark, CA, USA). The preparations were analyzed using a DuoScan-Meta LSM510 confocal laser scanning microscope (Carl Zeiss Meditec AG, Jena, Germany) equipped with a Plan-Apochromat 63 × 1.40 (numerical aperture) oil Ph3 objective.

### 2.8. Sucrose Gradient

Ten T-75 flasks of wild-type HEK293 cells were cultured under standard conditions, harvested at a sub confluent density, frozen in liquid nitrogen, and stored at −80 °C until use. The cell pellet was thawed on ice and underwent nucleus extraction according to the protocol [16]. Purified nuclei were treated with SN2 buffer supplemented with DNAse I to disrupt the nuclear membrane and destabilize the nucleolus following the PSE method [17]. Lysates were centrifuged (12,300 × *g*, 10 min, 4 °C), and supernatants were loaded on linear 15–40% sucrose gradient containing 20 mM of HEPES-NaOH (pH 7.5), 200 mM of NaCl, 4 mM of EDTA, 0.1% Igepal CA-630, 0.1 mg/mL of heparin, and 1 mM of DTT. Ultracentrifugation was performed with a SW40Ti rotor (Beckman Coulter, Brea, CA, USA) at 36,000 × rpm for 6 h at 4 °C. Fractions (0.5 mL) were collected on a gradient collector system (Bio-Rad Laboratories, Hercules, CA, USA) and subsequently subjected to Western and Northern blot analyses. Samples were extracted through the phenol:chloroform:isoamyl alcohol mix. Aqueous phases were collected for RNA extraction, while organic phases were used for protein precipitation. Northern and Western blots were performed according to the corresponding section in Materials and Methods.

### 2.9. Polysome Profiling

Three T-75 flasks of each cell line with RPF1/ESF1 knockdown as well as control cells were cultured to obtain approximately 70–80% confluence. Cycloheximide was added to each flask to obtain the final concentration of 100 μg/mL, and cells were incubated under normal conditions for 30 min. After that, flasks were removed from the CO_2_ incubator and immediately placed on ice. Culturing media were completely aspirated, and cells were rinsed with 10 mL of ice-cold PBS+cycloheximide (100 μg/mL). A total of 500 μL of the polysome lysis buffer (20 mM of Tris-HCl, 250 mM of NaCl, 15 mM of MgCl_2_, 1 mM of DTT, 0.5% Triton X-100, cycloheximide at 100 μg/mL, 20 U/mL of DNAse I, RNAse inhibitor at 1 U/μL, at pH 7.5) were added to each flask and cells were scraped. The obtained lysates were transferred to 2 mL tubes, incubated for 20 min on ice, and centrifuged for 10 min at a maximum speed of 4 °C. Their concentration was measured, and 15 OD (approximately 350 μL) of each sample were loaded on a linear 10–60% sucrose gradient (the sucrose gradient buffer contains 20 mM of Tris-HCl, 250 mM of NaCl, 15 mM of MgCl_2_, at pH 7.5). Samples were centrifuged for 3 h at 35,000 rpm at 4 °C. After centrifugation, gradients were manually fractionated from the top to the bottom, and the optical density of each sample was measured at 260 nm using a NanoDrop OneC spectrophotometer (ThermoFisher, Wilmington, DE, USA).

### 2.10. Ethynyl Uridine (EU) Pulse Labeling

Stable cell lines with shRNA #1/#2 against *RPF1* mRNA as well as control cells were grown to 70% confluence in 6-well plates. EU (Jena Bioscience, Jena, Germany) was added to the final concentration of 0.5 mM, and cells were incubated for 1 h. Total RNA was extracted using the TRIzol reagent (ThermoFisher, Wilmington, DE, USA). A total of 5 μg of RNA were incubated in the click reaction buffer (final volume of 50 μL; it contains biotin-azide (PEG4 carboxamide-6-Azidohexanyl Biotin, ThermoFisher, Wilmington, DE, USA), CuSO_4_, THPTA, sodium ascorbate, RNAse inhibitor 40 U, MOPS, EDTA, DEPC-treated water, at pH 7) at 25 °C, with 600 rpm in a thermo-shaker for 45 min. Then, 1 μL of RNAse-free glycogen (20 μg/μL, New England Biolabs, Ipswich, MA, USA, B1564S), 50 μL of 3 M of sodium acetate (pH 5.2), and 700 μL of cold 96% ethanol were added to the reaction mix. Samples were incubated overnight at −80 °C. RNA was pelleted by centrifugation at 13,000 × *g* for 20 min at 4 °C. The supernatants were removed, and RNA pellets were rinsed with 75% ethanol twice by centrifuging at 13,000× *g* for 5 min at 4 °C. Pellets were air-dried for 5 min at room temperature and resuspended in 10 μL of RNAse-free water. RNA integrity was checked in formaldehyde agarose gel (see Appendix A).

### 2.11. Extraction of Biotinylated RNA Using Streptavidin Magnetic Beads

A total of 1 μg of biotinylated RNA isolated following EU labeling was used for binding to 50 μL of streptavidin magnetic beads (ThermoFisher, Wilmington, DE, USA, 88816) equilibrated with the BW buffer prior to incubation with RNA (bind and wash buffer, and the wash step was repeated 3 times; the buffer composition was as follows: 10 mM of Tris-HCl, 1mM of EDTA, 2 mM of NaCl, at pH 7.5). Then, RNA was mixed with 125 μL of 2× bind and wash buffer, 2 μL of RNAse inhibitor, and RNAse-free water to reach a final volume of 250 μL. The mix was heated at 68 °C for 5 min and added to 50 μL of pre-equilibrated streptavidin magnetic beads. The suspension was incubated at 25 °C in a thermo-shaker for 45 min at 900 rpm. Then, the beads were sequentially washed 5 times with the 10× bead volume of the bind and wash buffer and 5 times with the 10× beads volume of the low-salt buffer (0.1 M of NaCl). Finally, the beads were resuspended in 50 μL of the low-salt buffer for cDNA synthesis.

### 2.12. Image Processing and Quantification

NPM1, RPF1, or SURF6 signals were quantified using a custom Python 3 script. Images were preprocessed with ImageJ to convert the DAPI and NPM1 channels into separate 16-bit grayscale images. A minimum of 56 cells were analyzed for each treatment group. Nuclei segmentation was determined in the DAPI images using Li thresholding functions in the Scikit-Image Python package. The CV for individual nuclei, defined as a standard deviation of pixel intensity divided by mean pixel intensity, was calculated from the NPM1 images using the SciPy Python package. All data were normalized to the scramble control in each experiment. Data are represented as box plots with whiskers generated by the Matplotlib and Seaborn Python packages.

### 2.13. AlamarBlue Assay

The stable cell lines were seeded in 96-well plates at 50% confluence. A total of 10% of the AlamarBlue reagent was added according to the manufacturer’s protocol (ThermoFisher, Wilmington, DE, USA) to a culture medium, and cells were incubated for 4 h at 37 °C, 5% CO_2_. Absorbance was measured at 570 nm using Thermo Scientific Multiskan EX ELISA reader (ThermoFisher, Wilmington, DE, USA). Experiments were performed in triplicates.

### 2.14. Statistical Analysis

Results are expressed as mean ± standard error of mean (SEM). Microsoft Excel and GraphPad Prism 9.0 software were used for statistical analysis of data.

## 3. Results

### 3.1. Knockdown of RPF1 or ESF1 Does Not Disrupt Gross Nucleolar Morphology but Induces Nucleophosmin Translocation/Accumulation in the Nucleoplasm

RPF1 or ESF1 knockdown in HEK293 cells was performed by lentiviral transduction with vectors containing the shRNA-expressing cassettes. For each protein, two different shRNAs were designed and tested to rule out any potential off-target effects. The levels of the remaining proteins and the corresponding mRNAs were analyzed by Western blotting and real-time PCR following antibiotic selection of the stably transduced cells. *RPF1* mRNA was decreased by 87% and 78% for shRNA #1 and shRNA #2, respectively (Appendix A). The RPF1 protein level was decreased by 53% and 52%, respectively (Figure 2a). The *ESF1* mRNA content dropped by 64% and 72% for shRNA #1 and shRNA #2, respectively (Appendix A), whereas its protein level reduced by 77% (Figure 2c). While ESF1 knockdown was efficient, the RPF1 protein level reduced only two-fold despite a dramatic decrease in the *RPF1* mRNA. This might be caused by the counterselection resulting in protein stabilization in the cells with stable RPF1 knockdown. Therefore, we switched towards a transient transfection with siRNA, which is supposed to prevent cell adaptation to RPF1 loss. As can be seen (Figure 2b, Appendix A), siRNA-mediated transient knockdown of RPF1 was efficient at both the mRNA and protein levels (90% or higher knockdown efficiency).

The depletion of ribosome biogenesis factors may induce nucleolar stress and the disruption of the nucleolar structure due to an imbalance in pre-ribosome assembly. To examine potential changes in the nucleolar architecture, we co-stained the nucleoli in HEK293 with the scramble shRNA, stable knockdown of either RPF1 or ESF1 using anti-NPM1 (also known as nucleophosmin or nucleolar phosphoprotein B23, the granular component marker protein), anti-SURF6 (co-localizes with NPM1 in the granular component according to our data [18]), anti-RPF1, and anti-ESF1 polyclonal antibodies. We observed the translocation of NPM1 but not SURF6 in the case of RPF1 and ESF knockdown, which is in contrast with the control cells. The extent of NPM1 or SURF6 redistribution in the cells with RPF1 and ESF1 knockdown was calculated by determining the coefficient of variation (CV) of the NPM1 or SURF6 pixel intensities within the nucleolus of each cell. These CV values were normalized to the average CV of the scramble control, and the data for an individual cell were plotted in parallel. A lower CV value indicates the exit from the nucleolus and the spreading of NPM1 or SURF6 throughout the nucleoplasm. Confocal images showed dramatic NPM1 translocation/accumulation in the nucleoplasm in the case of RPF1 knockdown, whereas SURF6 did not leave the granular component (Figure 3a–c, Appendix A). ESF1 knockdown induced a similar though less pronounced redistribution of NPM1; however, SURF6 remained in the granular component (Figure 4a–c, Appendix A).

Previously, SURF6 was shown to form a highly extended molecular network with other nucleolar proteins and nascent ribosomal RNAs through weak multivalent interactions. SURF6 has been suggested to regulate the nucleolar composition and biophysical properties as well as to dynamically modulate the nucleolar scaffolding gradient that guides the path of the ribosomal particle assembly [19]. Together with our results, these data indicate that although RPF1 or ESF1 deficiency induces NPM1 translocation, the gross nucleolar morphology is not disrupted. Simultaneously, pre-ribosomal subunit assembly sites may undergo certain changes leading to NPM1 dissociation and its exit into the nucleoplasm.

In addition, the localization of the remaining RPF1 or ESF1 was determined by microscopy. Despite the 50% efficiency of RPF1 knockdown according to Western blotting, weak trace signals of RPF1 were predominantly detected in the nucleoplasm (Figure 3b). The same was true for ESF1, which demonstrated similar localization in ESF1-deficient cells (Figure 4b). These observations allow us to speculate that RPF1 and ESF1 translocation from the nucleolus to the nucleus in the case of their knockdown might provide an additional explanation of their effects on ribosome biogenesis.

### 3.2. Knockdown of RPF1 Increases 5′ETS Abundance

NPM1 is the most abundant nucleolar protein exerting multiple functions. It shuttles between the nucleolus and the nucleoplasm and, in some cases, can leave the nucleus and enter the cytoplasm. Furthermore, NPM1 may play the role of a histone chaperone in the nucleolus and mediate the chromatin dynamics of ribosomal DNA. NPM1 accumulation in the nucleoplasm happens as a result of exposure to extra- and intracellular stresses, such as serum or glucose depletion, the toxic effects of anticancer drugs (e.g., doxorubicin, tiazofurin), ribosomal transcription inhibition (e.g., Actinomycin D), and others [20,21,22]. As seen in Figure 3, RPF1 knockdown leads to a remarkably strong accumulation of NPM1 in the nucleoplasm, thereby indicating possible effects on Pol I transcription. To test this hypothesis, we performed the pulse labeling of scramble control HEK293 and cells with shRNA-mediated RPF1 knockdown with EU followed by click biotinylation and the isolation of biotin-labeled RNA, and we quantified the relative amount of the 5′ETS precursor. The data demonstrate that the earliest short-lived 5′ETS precursor was markedly elevated in the cells with RPF1 knockdown (Appendix A).

### 3.3. RPF1 Knockdown Predominantly Leads to the Alteration of the Profile of Pre-60S rRNAs

As mentioned above, the structural and proteomic characteristics of the nucleolus in higher eukaryotes imply the existence of additional pre-ribosomal maturation steps as well as currently unknown functions of the human homologs of yeast proteins. To evaluate the possible role of RPF1 in human ribosome biogenesis, we performed Northern blotting hybridization and RAMP (Ratio Analysis of Multiple Precursors) analysis of the total RNA samples obtained from control HEK293 cells and RPF1-deficient cells.

RPF1 depletion by shRNA resulted in a slight decrease in the primary 45S/47S pre-rRNA that could be detected by hybridization with an ITS1-specific probe (Appendix A). At the same time, we also observed the accumulation of 32S, 30S, and 26S as well (Figure 1, Appendix A). These changes in the stationary levels of pre-rRNAs could often arise from several molecular events occurring at different stages of pre-rRNA processing. A detailed study of the molecular events leading to the accumulation/decrease in pre-rRNAs requires a nuanced analysis of the ratios between several different precursors: the RAMP analysis [15]. We assessed the band intensities by Northern blotting, and the data are presented as RAMP diagrams. We found that RPF1 knockdown elevated the ratio of 32S, 30S, and 26S to the 47S precursor as well as of the 32S to 41S precursor. Moreover, 12S/32S and 26S/30S ratios were reduced (Figure 5). Changes in the levels of 32S/47S, 32S/41S, and 12S/32S indicated the inhibition of the 32S precursor processing at site 4 within the ITS2 region (see the diagram in Figure 1; it corresponds to the C2 site in yeast [23]). The increased ratios observed for the 30S and 26S precursors to the primary 47S transcript may be related to the slight decline in the 47S precursor level in the RPF1-deficient cells or to the impaired processing within the 5′ETS precursor.

The effects of siRNA-mediated knockdown of RPF1 were evaluated as well. The 15 nM siRNA #2 provided the most pronounced decrease in the RPF1 mRNA and protein levels (Figure 2b, Appendix A). According to the RAMP analysis results, the overall effect of siRNA knockdown of RPF1 was less profound compared with that of shRNA knockdown. Furthermore, we observed a decrease in the 12S/32S, 12S/47S, 32S/41S, and 30S/47S levels as well as a slight increase in the 26S/30S ratio (Figure 1 and Figure 6). Although these effects differ from those observed for stable cell lines, they all indicate changes in the 41S–32S–12S axis of ribosome biogenesis. These rather inconsistent effects of the stable and transient knockdown of RPF1 may be explained through the adaptation of cells to the RPF1 deficiency, leading to the activation of alternative mechanisms of rRNA processing and ribosome assembly. In the case of transient knockdown, the period of time seems to be insufficient for selecting cells with compensatory changes. Thus, transient knockdown is likely to reveal the stages involving RPF1 in wild-type HEK293.

### 3.4. ESF1 Knockdown Induces Changes in the Profile of Pre-40S rRNAs and a Shift in the Biogenesis Pathways

ESF1 knockdown exerted several effects on the pre-rRNA profile, either affecting the pre-40S processing or switching the biogenesis pathways. First, we observed the 30S rRNA precursor accumulation and no changes in the 47S rRNA level (Figure 1, Figure 7a and Appendix A). This resulted in an elevation in the 30S/47S and a decrease in the 26S/30S ratios (Figure 7b). Taken together, these data provide solid evidence for the inhibition of the pre-rRNA cleavage predominantly at the A0 site within 5′ETS (corresponding to the A0 site in yeast [24]). In yeast, Esf1 loss inhibits pre-rRNA processing not only at the A0 site, but also at the A2 site (corresponding to the E site in humans [23]). Nevertheless, our data indicate that the inhibitory effect of ESF1 knockdown on the E site in human cells is not significant.

Second, we observed the elevated 32S and decreased 41S levels (Figure 1, Figure 7a and Appendix A). RAMP analysis revealed an increase in the 32S/47S, 32S/41S, and 21S/41S ratios as well as a decrease in the 41S/47S ratio (Figure 7b). These results indicate the acceleration of pathway 2 processing, which involved alternative stages required either for the cleavage at the A0 site or for skipping it to maintain the level of mature 18S (Figure 1).

### 3.5. RPF1 and ESF1 Co-Sediment with Precursors of 60S and 40S Subunits, Respectively

Downmodulating the key proteins by shRNA-mediated knockdown can have indirect effects on intracellular processes. Therefore, to prove that RPF1 and ESF1 are directly involved in ribosome biogenesis, it is necessary to demonstrate their interaction with the pre-60S and pre-40S ribosome subunits, respectively. To confirm this, sucrose gradient centrifugation experiments were performed. RNA and protein samples isolated from each fraction were analyzed by Northern and Western blotting followed by hybridization with oligonucleotides or immunodetection. Analysis of the Northern blotting hybridization with the specific probes for 28S and 18S rRNA demonstrated the sufficient separation of the 60S and 40S precursors (Figure 8b). Hybridization with the ITS2 probe revealed the enrichment of the loaded sample with the ribosomal precursors. The obtained results demonstrate the presence of the pre-ribosomal particles containing the 32S and 12S precursors, implying the efficient removal of cytoplasmic mature subunits (Figure 8b). Western blots with anti-RPF1 and anti-ESF1 antibodies revealed the co-sedimentation of RPF1 with the pre-60S particle (Figure 8a), while ESF1 co-sedimented with the pre-40S precursor (Figure 8a).

### 3.6. Cells with RPF1 and ESF1 Knockdown Have Polysome Profiles Similar to Wild-Type Control Ones

The depletion of ribosome biogenesis factors should theoretically affect the assembly of the corresponding subunits and result in a decrease in the polysome fraction due to insufficient formation of translation-competent ribosomes. To check these possibilities, the polysome fraction gradients were performed using cell lysates obtained from HEK293 cells expressing shRNAs against *RPF1* or *ESF1* mRNAs, while control HEK293 cells with scramble shRNA served as a control.

RPF1 knockdown did not result in any detectable changes in the 60S–80S peak as well as the polysome peaks (Figure 9a). As for ESF1 knockdown, it caused a detectable decrease in the 40S peak and did not affect the polysome peaks (Figure 9b). These findings imply that cells with RPF1 or ESF1 depletion retained their translation activity, which was required for maintaining the bulk protein synthesis.

## 4. Discussion

Ribosome biogenesis is a well-orchestrated process involving myriads of assembly factors and non-coding RNAs that modulate the pre-rRNA processing. Progressing along the pathway is tightly regulated by the extra- and intracellular conditions to maintain an optimal protein synthesis routine.

The formation of the mature 28S ribosomal RNA during ribosome biogenesis in higher eukaryotes requires two major events. First, the RNAse MRP mediates the cleavage of the 41S or 45S pre-rRNA at site 2 (corresponding to the A3 site in *S. cerevisiae)* in the 21S and 32S pre-rRNA intermediates in pathways 1 and 2, respectively [25]. LAS1 endonuclease in the Rixosome complex (NOL9, WDR18, LAS1L, MDN1, PELP1, TEX10, and SENP3) performs the second cleavage at site 4 (corresponding to site C2 in *S. cerevisiae* [19]). This promotes the formation of the 12S and 28S pre-rRNA intermediates in both of the biogenesis pathways leading to NOG2-NSA2 dissociation and PET maturation [26,27]. The formation of a favorable structural environment for rRNA domain I precedes the cleavage event at this site [28]. The early nucleolar SSF1 particle interacts with the so-called WDR74-module (WDR74-NOP52-RPF1-MAK16, corresponding to the Nsa1-module in *S. cerevisiae)* [29]. This complex clamps down domains I and II, which mark out the future PET site, and forms the intermediate NSA1 nucleolar precursor. During the subsequent maturation steps, AAA-ATPase NLV2 (corresponding to Rix7 in *S. cerevisiae*) is recruited by promoting WDR74-module release, which is coupled with or precedes the dissociation of the SSF1-RRP15-SURF6 complex [30]. These events lead to the formation of the PET rim and vestibule [12,22,23,24,25,26,27,28]. This is followed by the association of the 60S precursor with the forming factors of domain III and the subsequent recruitment of AAA-ATPase MDN1 (corresponding to Rea1 in *S. cerevisiae*), which stimulates the detachment of BOP1-WDR12-DDX18 (corresponding to Erb1, Ytm1, and Has1 in *S. cerevisiae*, respectively) and related factors [31]. These processes are coupled with the incorporation of the NOG2 and 5S RNP complex and the formation of the late nucleolar 60S precursor [12,28,32,33]. The NOG2 particle interacts with the Rixosome complex, which possesses nuclease activity; it mediates ITS2 cleavage, PET, and central protuberance formation after its dissociation [12,27,28].

How do our functional and biochemical data on RPF1 fit into this complex picture? Although RPF1 depletion downmodulated site 4 cleavage, the effect of RPF1 knockdown was indirect. This idea is supported by the fact that the Rixosome complex did not contain RPF1. Cryo-EM data suggest that RPF1 is no longer associated with the 60S precursor at the time of site 4 cleavage. It is likely that the structural support of domains I and II provided by the WRD74 module was not sufficient due to PRF1 absence in the complexes formed prior to site 4 cleavage. We suggest that the lack of RPF1 might negatively affect the formation of Rixosome- or LAS1-binding sites and results in the lowered efficiency of 32S pre-rRNA cleavage at site 4 within the ITS2 region. Nevertheless, this defect in ITS2 grounding and cleavage is insufficient for the significant disruption of the ribosome biogenesis and cell proliferation rate, which is supported by the results of AlamarBlue assay (Appendix A). The same results have been observed for *S. cerevisiae.* Pre-60S particles can bypass the nuclear quality control mechanism, enter the following maturation steps in the cytoplasm, and interact with the mature 40S subunit. Indeed, these aberrant 80S ribosomes can be found in the polysome fraction. These observations are supported by our polysome profiling data that show that 60S-80S and polysome fractions are not depleted in RPF1-deficient cells. However, the mechanisms of the quality control evasion of the ITS2-containing particles and the translation properties of these abnormal ribosomes still remain elusive [34].

Of note, 5′ETS augmentation in cells with shRNA-induced RPF1 knockdown suggests either an increase in the Pol I transcription rate or inhibited 5′ETS cleavage. According to the published data, NPM1 absence in the nucleolus may lead to a decrease in H3K9 methylation at the promoters of the rDNA genes [21]. We observed the disruption of the 41S–32S–21S axis in the stable cell lines with RPF1 knockdown. An elevated level of the 5′ETS precursor also indicated that RPF1 might contribute to the early steps of ribosome biogenesis, and its absence resulted in the deregulation of A0 processing. This assumption is in line with the data obtained from yeast, which revealed that RPF1 co-immunoprecipitated with an early 35S rRNA-containing precursor. Eventually, an elevated 5′ETS level might be caused by either rDNA promoter demethylation or decreased 5′ETS processing as well as both these factors.

18S rRNA maturation in humans requires more advanced re-modeling and processing steps in the nucleoplasm compared with the 18S formation in yeast [28,35,36]. The human 47S primary transcript contains a 5′ETS region that is six times longer [37]. The early stages of human pre-ribosomal RNA processing appear to begin with the association of a nascent transcript with UTP-A, UTP-B, and U3 snoRNA (similar to yeast) to stabilize the 5′ETS segment, form the 5′ETS particle, and provide the favorable structural conditions for the docking of the other assembly factors [28,38]. The modular assembly of the 5′-, central, 3′-major, and 3′-minor domains of the 18S rRNA segment starts only when it becomes accessible. The proper domain organization and association with the assembled 5′ETS complex leads to cleavage at the A0 site by the endonuclease UTP24 both in yeast and in humans. Subsequent complex re-modeling is accomplished by the association of new assembly factors and SSU processome formation [5,28]. The simplicity of yeast nucleolar organization allowed for isolating and describing the stepwise domain formation by Cryo-EM. These studies resolved a series of 5′ETS structures in different states [2,3,4,5,7]. The 5′ETS formation in humans remains a mystery due to a lack of structural data [39]. The ESF1 protein and its yeast homolog Esf1 are the earliest assembly factors. They interact with the 5′ domain of the 18S rRNA segment during SSU processome formation according to the biochemical data [5,7]. Despite significant progress in the structural characterization of pre-SSU, 3D models of yeast particles do not provide any information regarding the 5′ domain of 18S with the attached Esf1 protein and its partners. Earlier, it was shown that while each 18S rRNA domain predominantly grows as an independent module, the presence of these domains in favorable spatial organization leads to A0 site cleavage. This is accompanied through the almost immediate dissociation of the earliest factors and the compaction of the precursor into SSU [5]. According to our data, ESF1 knockdown resulted in the 41S/47S and 26S/30S ratio decline, thereby strongly demonstrating a reduced efficiency in the A0 site cleavage. Reduced A0 site processing resulted in an elevation in the 32S/47S and 30S/47S ratios. This finding may serve as solid evidence for pathway 2 activation, which starts with site 2 cleavage instead of that of A0. The 26S/30S ratio was drastically reduced; however, the net level of matured 18S did not significantly change. This implies that the A0 site cleavage is most likely skipped during rRNA maturation along pathway 2.

Together with our previous data [18], these results demonstrate that human cells utilize ribosome biogenesis pathways extremely effectively to maintain a bulk protein synthesis level. Knockdown of the pre-ribosome assembly factors, especially those that exert accessory functions and are not directly involved in cleavage events, did not drastically drop the rate of ribosomal subunit maturation but rather fine-tuned the maturation events and the re-direction of the pre-rRNA processing paths. These assumptions are supported by our polysome profiling data. Finally, it should be noted that for our study, we utilized transformed cells that likely possess altered control mechanisms, which might result in increased plasticity and the relaxed control of ribosome biogenesis.

## Figures and Tables

**Figure 1 cells-13-00326-f001:**
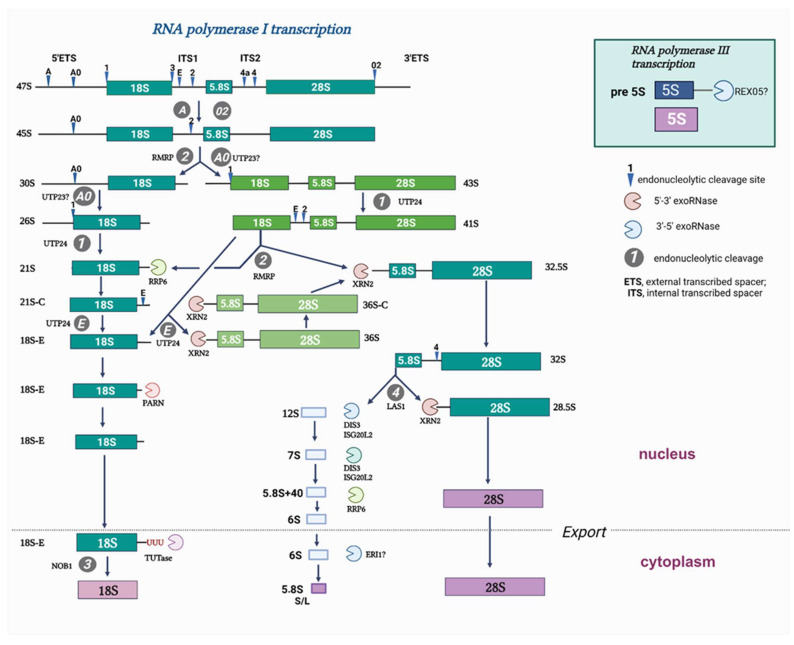
Pre-rRNA processing pathways in mammalian cells.

**Figure 2 cells-13-00326-f002:**
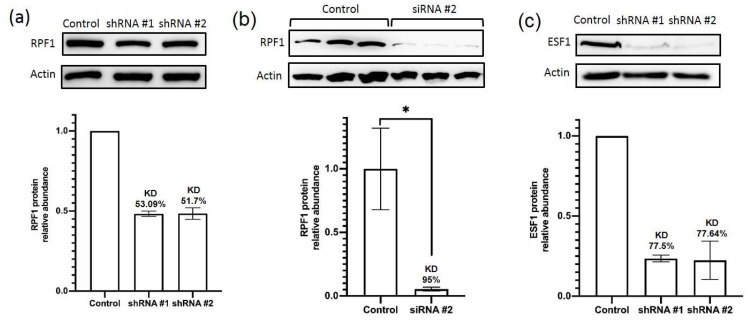
shRNA- and siRNA-mediated knockdown of RPF1 and ESF1 in HEK293 cells. (**a**) Western blot analysis of lysates obtained from the cells stably transduced by the scramble (control) or shRNAs against *RPF1* mRNA (shRNA #1 and shRNA #2). Immunostaining of PAAG-separated proteins transferred to a membrane and probed with anti-RPF1 IgG is shown. Actin was used for normalization. Mean values of two independent experiments are shown. (**b**) Western blot analysis of lysates obtained from the cells transfected by the scramble (control) siRNA or siRNA #2 against *RPF1* mRNA. Immunostaining of PAAG-separated proteins transferred to a membrane and probed with anti-RPF1 IgG is shown. Actin was used for normalization. Mean values of three independent experiments are shown. Star indicates statistically significant differences between control and cells with siRNA-mediated knockdown of RPF1 (* *p* < 0.05). (**c**) Western blot analysis of lysates obtained from the cells stably transduced by the scramble (control) or shRNAs against *ESF1* mRNA (shRNA #1 and shRNA #2). Staining of PAAG-separated proteins transferred to a membrane and probed with anti-ESF1 IgG is shown. Actin was used for normalization. Mean values of two independent experiments are shown.

**Figure 3 cells-13-00326-f003:**
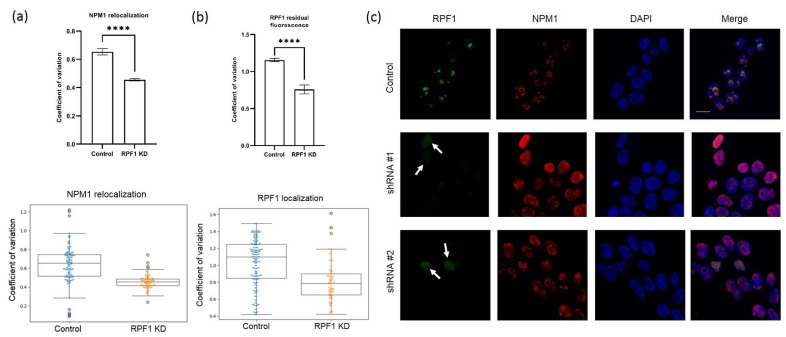
Relocation of NPM1 from the nucleoli to the nucleoplasm in HEK293 with RPF1 shRNA-mediated knockdown. (**a**) Quantifications of NPM1 in different parts of the nucleus (nucleoli vs. nucleoplasm) of cells with stably decreased RPF1 (shRNA-expressing constructs were integrated into the genome) relative to control cells. Coefficients of variation were calculated and plotted as boxplots with whiskers, as described in the Materials and Methods section. (**b**) Quantifications of the residual RPF1 signal in the nucleoplasm of cells with stably decreased RPF1 relative to the control cells. Coefficients of variation were calculated and plotted as boxplots with whiskers, as described in the Materials and Methods section. (**c**) Confocal images of HEK293 cells stably transduced by the scramble (control) or shRNAs against *RPF1* mRNA (shRNA #1 and shRNA #2). Cells were grown on cover slides and fixed, permeabilized, and stained with the antibodies against NPM1, RPF1. DAPI was used to visualize the nuclei. Arrows indicate the residual signals from the RPF1 protein in the nucleoplasm. Stars indicate statistically significant differences between NPM1 localization and RPF1 residual fluorescence in nucleoli of control and cells with knockdown of RPF1 (**** *p* < 0.0001).

**Figure 4 cells-13-00326-f004:**
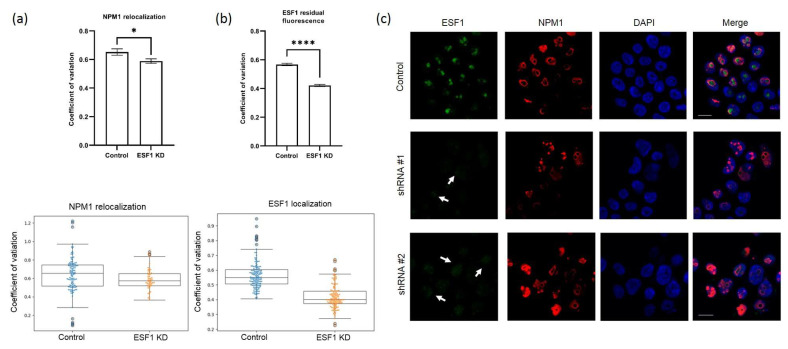
Relocation of NPM1 from the nucleoli to the nucleoplasm in HEK293 with ESF1 shRNA-mediated knockdown. (**a**) Quantifications of NPM1 in different parts of the nucleus (nucleoli vs. nucleoplasm) of cells with stably decreased ESF1 (shRNA-expressing constructs were integrated into the genome) relative to control cells. Coefficients of variation were calculated and plotted as boxplots with whiskers, as described in the Materials and Methods section. (**b**) Quantifications of the residual ESF1 signal in the nucleoplasm of cells with stably decreased ESF1 relative to control cells. Coefficients of variation were calculated and plotted as boxplots with whiskers, as described in the Materials and Methods section. (**c**) Confocal images of HEK293 cells stably transduced by the scramble (control) or shRNAs against *ESF1* mRNA (shRNA #1 and shRNA #2). Cells were grown on cover slides and fixed, permeabilized, and stained with the antibodies against NPM1, ESF1. DAPI was used to visualize the nuclei. Arrows indicate the residual signals from the ESF1 protein in the nucleoplasm. Stars indicate statistically significant differences between NPM1 localization and ESF1 residual fluorescence in nucleoli of control and cells with knockdown of ESF1 (* *p* < 0.05; **** *p* < 0.0001).

**Figure 5 cells-13-00326-f005:**
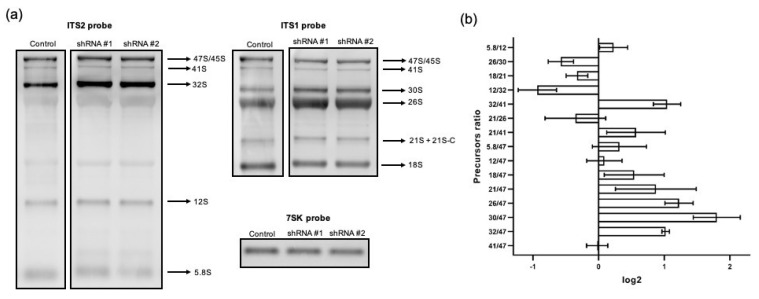
Changes in the pre-rRNA processing in HEK293 with shRNA-mediated RPF1 knockdown. (**a**) Agarose gel separated the samples of total RNA from control HEK293 and cells stably expressing shRNAs against *RPF1* mRNA, which were transferred to the membrane and probed with biotinylated oligonucleotide probes used for ITS1, ITS2, and 7SK RNA (used as a loading control) vizualisation. One representative experiment of the three is shown. (**b**) Results of the RAMP analysis of the images from the left (for details, see Materials and Methods and main text). Plots show normalized mean values of ratios between different rRNA precursors.

**Figure 6 cells-13-00326-f006:**
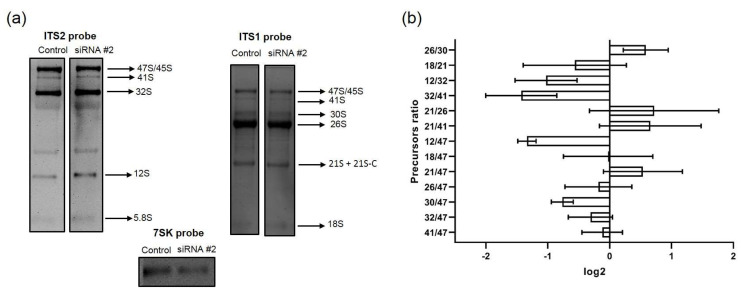
Changes in pre-rRNA processing in HEK293 with siRNA-mediated RPF1 knockdown. (**a**) Agarose gel separated the samples of total RNA from control scramble HEK293 and cells, which were transiently transfected with siRNA against *RPF1* mRNA and were transferred to the membrane and probed with biotinylated oligonucleotide probes used for ITS1, ITS2, and 7SK RNA (used as a loading control) vizualisation. One representative experiment out of three independent experiments is shown. (**b**) Results of the RAMP analysis of images from the left (for details, see Materials and Methods and the main text). Plots show normalized mean values of ratios between different rRNA precursors. The experiment was repeated three times.

**Figure 7 cells-13-00326-f007:**
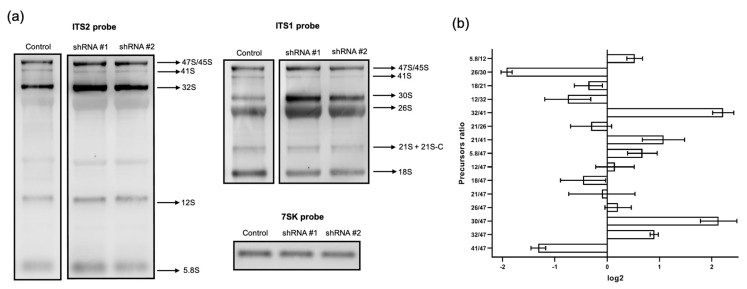
Changes in pre-rRNA processing in HEK293 with shRNA-mediated ESF1 knockdown. (**a**) Total RNA samples, isolated from control HEK293 with the scramble shRNA and cells stably expressing shRNAs against *ESF1* mRNA, which were separated in agarose gel, transferred to the membrane, and probed with biotinylated oligonucleotide probes used for ITS1, ITS2, and 7SK RNA (serving as a loading control) vizualisation. A representative image showing the result of one out of the three experiments is shown. (**b**) Results of RAMP analysis of the images from the left (for details, see Materials and Methods and the main text). Plots show normalized mean values of the ratios between different rRNA precursors.

**Figure 8 cells-13-00326-f008:**
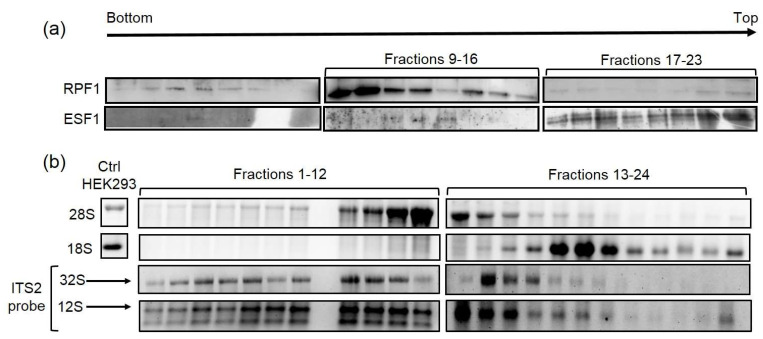
RPF1 and ESF1 were detected in the fractions of a sucrose gradient containing the pre-60S and pre-40S particles, respectively. The nucleoli isolated from HEK293 cells were separated using a sucrose gradient. Proteins and RNA were isolated from the individual fractions of the gradient. (**a**) Protein extracts were analyzed by Western blotting and stained with the anti-RPF1 and anti-ESF1 IgG. (**b**) RNA samples from individual fractions were analyzed by Northern blotting and stained with the probes used for 28S, 18S rRNA and ITS2 vizualisation.

**Figure 9 cells-13-00326-f009:**
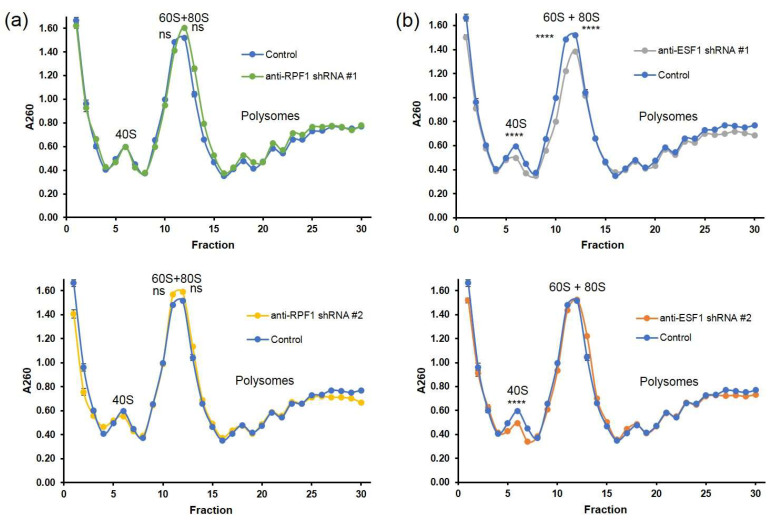
Polysome profiles of the cytoplasmic fraction from control cells or cells with RPF1 or ESF1 knockdown. Sucrose gradient profiles were obtained from cells with either shRNA-mediated RFP1 (**a**) or ESF1 (**b**) knockdown and from control cells. Fractions were collected manually, and A260 was measured using a plate reader spectrophotometer. Stars indicate statistically significant differences between 40S and 60S+80S peaks in control and cells with knockdown of ESF1 (**** *p* < 0.0001).

## Data Availability

Data is contained within the article or Appendix A.

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
