# Peer review of "Human RPF1 and ESF1 in Pre-rRNA Processing and the Assembly of Pre-Ribosomal Particles: A Functional Study"

_cells, 2024, doi:10.3390/cells13040326_

Round 1

Reviewer 1 Report

Comments and Suggestions for Authors

In this study, the authors investigated roles of human RPF1 and ESF1 in ribosome biogenesis, and showed that RPF1 and ESF1 are involved in the pre-rRNA processing in the pre-60S and pre-40S particles, respectively.

My main concern is the knockdown efficiency of RPF1.

a) According to the authors, the knockdown efficiency was around 50% at the protein level; however, it is hard for me to believe the data of quantification as the band corresponding to RPF1 remained clearly in the RPF1 shRNA #1 and shRNA #2.

b) Although the authors mentioned a decrease in RPF1 mRNAs as determined by RT-qPCR but there was no data.

c) In Figure 1c, virtually there were no signals of RPF1 in the RPF1 shRNA #1 and shRNA #2 cell lines. However, ~50% of RPF1 protein should remain according to Figure 1a and 1b. How is the data interpreted? If the amount of RPF1 was decreased by 50% in each cell, the fluorescent signal of RPF1 in the shRNA cell lines should be 50% compared to the control cells. If RPF1 was totally knock-downed in the 50% of population, then the fluorescent signal of RPF1 in 50% of shRNA cell lines should be zero and, in the other 50% of shRNA cell lines, the signal should be comparable to the one in the control cells.

d) Under this condition where RPF1 was slightly reduced, in Figure 6, the authors detected very modest changes in the pre-rRNA processing although RMAP diagrams were shown.

The knockdown experiment is very important to investigate the function of those proteins and therefore I suggest the authors use other knockdown approaches such as siRNA-based knockdown and/ or so-called ‘rescue’ experiments using an siRNA/ shRNA resistant RPF1 expressing vector.

Another concerns are listed below:

*Re-localization of B23 is often detected when RNA polymerase (Pol) I is inhibited (e.g. PMID: 22871610, PMID: 22871610, PMID: 22871610 etc). Because the effect of knockdown of RPF1 on the pre-rRNA processing is modest, this protein might be involved in the Pol I transcription rather than the pre-rRNA processing. The authors should check this possibility.

*Figure 7: 1) there is no figure legend, 2) the authors just showed the separation of 28S and 18S rRNAs as a marker of pre-60S and pre-40S, respectively; however, because 28S and 18S rRNAs were detected as a marker, it is hard to distinguish pre-ribosomal particles from mature ribosomes, which are more abundant and easy to be contaminated. Therefore, the authors should show 1) western blotting of subcellular fractions obtained by the PSE method, and 2) northern blotting to detect pre-rRNA species contained in the pre-60S and pre-40S (e.g. 32S pre-rRNA for pre-60S and 21S pre-RNA for pre-40S).

*Figure 5a and 6a: ITS1 probe cannot detect 18S rRNA

*Figure 5c and 6c: AlamarBlue assay was used for cell viability and cell proliferation measurements but MTT assay was described in the method section

*Please mention in the Abstract and Introduction why the authors focus on RPF1 and ESF1 among many ribosome assembly factors

Comments on the Quality of English Language

Typos should be amended (RNAse, ESF11, etc…)

Author Response

First, my colleagues and I would like to thank you for careful and professional evaluation of our manuscript. We are pleased to receive kind remarks concerning the quality of data and scientific value of observations described in our study. The revised version of the manuscript was improved and not only shows the additional results on siRNA knockdown of RPF1, but also detailed confocal images analysis, polysome profiling experiments as well as ethynyl uridine labeling data for polymerase I transcription analysis. Main figures are presented in the text of the manuscript and additional PDF file with main and supplementary figures was prepared for the convenience. Our response contain more than one file, therefore we are not able to attach it through the standard form here. We would kindly ask you to use the link below to get the access to all files.

https://drive.google.com/drive/folders/1N5K4CsRsraFheU0wQVf0FtLAFmfN0k3p?usp=drive_link

  • cells-2562161_R1 – updated manuscript version;
  • Response #1 – file with answers to raised issues and concerns;
  • Updated figures – file with updated main and supplementary figures;
  • Excel files contain all raw data concerning RT-PCR, blots and graphs counting.

Reviewer 2 Report

Comments and Suggestions for Authors

This paper delves into a functional investigation of the roles played by human RPF1 and ESF1 proteins in ribosome biogenesis. It offers insights into their contributions to pre-rRNA processing and the assembly of pre-ribosomal particles. However, certain aspects of the paper require further clarity and refinement in order to enhance overall understanding and effectiveness. Here are some key comments and suggestions for improvement:

Line 202-206: The mRNA and protein levels as stated by the authors do not align with the figures presented. Clarification or corrections are needed to accurately reflect the data.

Figure 1: It would be helpful to clearly indicate that Figure 1b represents RT-PCR data. Additionally, the relative percentage of Figure 1a should be provided.

Figure 2: Ensure that the knockdown groups are properly labeled in the figure.

Figure 3: Similar to Figure 1, specify that Figure 3b represents RT-PCR data and include the relative percentage of Figure 3a.

Figure 7: A legend is missing for Figure 7. Additionally, authors need to add ribosome profiling peaks and labels to indicate the location of subunits.

Line 222-225: Provide a clear explanation for the contrasting behaviors of SURF6 and NPM1 in the granular component.

Line 226: The statement about ESF1 knockdown inducing similar but weaker redistribution of NPM1 is questionable due to the minimal differences observed in the results.

Line 227-230: The assertion that RPF1 or ESF1 depletion does not disrupt gross nucleolar morphology does not seem supported by the results, particularly concerning RPF1.

Line 275-277 & Line 299-301: The patterns of change in 30S and 26S levels resulting from knockdown of both RPF1 and ESF1 appear similar. As ESF1 function is discussed in relation to the A0 process, it could be considered whether RPF1 may have a similar role.

Line 327-339: This section lacks support from the results presented in Figure 7. It would be necessary to include ribosome profiling peaks for Free, 40S, 60S, 80S, and polysome. Additionally, the authors need to examine whether the loss or impaired function of these proteins influences their distributions in the gradient.

Discussion: Correct the reference to Utp24 instead of Utp23. Also, ensure proper references are included throughout this section where needed.

Comments on the Quality of English Language

Fine to read.

Author Response

First, my colleagues and I would like to thank you for careful and professional evaluation of our manuscript. We are pleased to receive kind remarks concerning the quality of data and scientific value of observations described in our study. The revised version of the manuscript was improved and not only shows the additional results on siRNA knockdown of RPF1, but also detailed confocal images analysis, polysome profiling experiments as well as ethynyl uridine labeling data for polymerase I transcription analysis. Main figures are presented in the text of the manuscript and additional PDF file with main and supplementary figures was prepared for the convenience. Our response contain more than one file, therefore we are not able to attach it through the standard form here. We would kindly ask you to use the link below to get the access to all files. https://drive.google.com/drive/folders/1jzJmzShEqcP2-2_lPSj4fUdesHwGA8LS?usp=drive_link

  • cells-2562161_R1 – updated manuscript version;
  • Response #2 – file with answers to raised issues and concerns;
  • Updated figures – file with updated main and supplementary figures;
  • Excel files contain all raw data concerning RT-PCR, blots and graphs counting.

Round 2

Reviewer 1 Report

Comments and Suggestions for Authors

I appreciate the additional siRNA-based approaches that the authors have performed and therefore the authors have addressed my concerns properly. However, additional data provided in this revised manuscript is a little confusing and therefore needs to be clarified.

I find it quite interesting that the EU-labeled RNA fragments in the 5’ETS region, as determined by RT-qPCR, were ~ 14-15 times more abundant in cells with RPF1 KD in comparison to control. Then, the authors discussed ‘A0 processing deregulation’. Since the authors established the methods of confocal microscopy and northern blotting, I would request additional and relatively easy experiments:

1)     To detect the EU-labeled nascent transcripts, the approach using fluorescent azides, followed by microscopic imaging, is frequently used compared to the method the authors used (please see the paper: https://www.pnas.org/doi/10.1073/pnas.0808480105 etc). The authors should perform this approach to check whether a fluorescence signal is also elevated in RPF1 KD cells.

2)     The authors should perform additional northern blotting with 5’ETS probe. Ideally, a probe sequence should be identical to the primer sequence to detect 5’ETS by RT-qPCR. This additional northern blotting would be useful because, if the processing dysregulation occurs, the authors should detect an accumulation of pre-rRNA precursors corresponding to 5’-A as well as 47/45S.

Moreover the followings are minor points that should be amended:

·        Line 110: Please include primer sequences used for RT-qPCR

·        Line 120: Please provide the source of primary antibodies and dilution for WB

·        Line 231: Please provide the data of ‘RNA integrity was checked in formaldehyde agarose gel.’

Author Response

    Firstly, my colleagues and I would like to sincerely thank you for your essential comments and suggestions during the reviewing process. We appreciate that you were mostly satisfied with our response, and raised issues and concerns were addressed properly. As well, we agree that independent and direct confirmation of increase in 5’ETS containing pre-ribosomal RNAs by northern blot and hybridization with corresponding probe would further prove our conclusions. At the same time, according to the previously published materials, RPF1 knockdown in S. cerevisiae leads to elevated 35S rRNA precursor (for reference: Wehner, K. A., & Baserga, S. J. The sigma (70)-like motif: a eukaryotic RNA binding domain unique to a superfamily of proteins required for ribosome biogenesis. Molecular cell, 9(2), 329–339. https://doi.org/10.1016/s1097-2765(02)00438-0). The results of yeast Rpf1 coimmunoprecipitation experiment can be found in the abovementioned manuscript.  They support knockdown data and suggest Rpf1/RPF1 participation in the earliest ribosomal RNA precursor processing but fail to provide molecular details. Our RT-PCR analysis data correlate well with the data obtained in S. cerevisiae. Anyway, we are interested in the obtained results and we will make additional efforts in our future works to analyze the role of RPF1 in the processing of the earliest ribosomal precursor.

    As for raised minor points, additional data were added as follows:

  • 5’ETS and GAPDH primers sequences (lanes 112-115 of the revised manuscript);
  • source and dilutions of primary antibodies (lanes 124-125 of the revised manuscript);
  • figure with RNA electrophoresis after click reaction (supplementary figure 11).

    Apart from mentioned above the previous version of the manuscript was revised by our native speaking colleague to improve language quality. The updated version is attached to this reply. As was done during the previous reviewing round, I added additional file with updated figures to the google disk and prepared the link. Please, follow it to view agarose gel with RNA after click reaction. https://drive.google.com/drive/folders/1N5K4CsRsraFheU0wQVf0FtLAFmfN0k3p?usp=sharing

    Thank you a lot in advance for the support and for the careful evaluation of our manuscript.

With sincerely best regards,

On behalf of co-authors,

Alexander Deryabin,

Research scientist,

Laboratory of molecular virology,

Institute of Bioorganic Chemistry RAS

Miklukho-Maklaya street 16/10, Moscow 117997, Russia

Tel.: +7 985 649 48 99

Reviewer 2 Report

Comments and Suggestions for Authors

My comments well addressed. 

Author Response

My colleagues and I would like to sincerely thank you for your valuable comments during the reviewing process.

With sincerely best regards,
On behalf of all co-authors,

Alexander Deryabin,
Research scientist,
Laboratory of molecular virology,
Institute of Bioorganic Chemistry RAS
Miklukho-Maklaya street 16/10, Moscow 117997, Russia
Tel.: +7 985 649 48 99
E-mail: [email protected]